# Use of Nanostructured Silica SBA-15 as an Oral Vaccine Adjuvant to Control *Mycoplasma hyopneumoniae* in Swine Production

**DOI:** 10.3390/ijms24076591

**Published:** 2023-04-01

**Authors:** Gabriel Y. Storino, Fernando A. M. Petri, Marina L. Mechler-Dreibi, Gabriel A. Aguiar, Leonardo T. Toledo, Laíza P. Arruda, Clarisse S. Malcher, Tereza S. Martins, Hélio J. Montassier, Osvaldo A. Sant’Anna, Márcia C. A. Fantini, Luís Guilherme de Oliveira

**Affiliations:** 1School of Agricultural and Veterinarian Sciences, São Paulo State University (Unesp), Jaboticabal 14884-900, SP, Brazil; 2Laboratório de Virologia Animal (LVA), Departamento de Veterinária, Universidade Federal de Viçosa, Avenida Peter Henry Rolfs s/n, Campus Universitário, Viçosa 36570-900, MG, Brazil; 3Department of Chemistry, Federal University of São Paulo (UNIFESP), Diadema 09913-030, SP, Brazil; 4Butantan Institute, São Paulo 05508-040, SP, Brazil; 5Physics Institute, University of São Paulo (USP), São Paulo 05508-090, SP, Brazil

**Keywords:** animal health, immunology, infectious diseases, respiratory diseases, vaccinology

## Abstract

*Mycoplasma hyopneumoniae* is a difficult-to-control bacterium since commercial vaccines do not prevent colonization and excretion. The present study aimed to evaluate the performance of an orally administered vaccine composed of antigens extracted from *Mycoplasma hyopneumoniae* and incorporated into mesoporous silica (SBA-15), which has an adjuvant-carrier function, aiming to potentiate the action of the commercial intramuscular vaccine. A total of 60 piglets were divided into four groups (n = 15) submitted to different vaccination protocols as follows, Group 1: oral SBA15 + commercial vaccine at 24 days after weaning, G2: oral vaccine on the third day of life + vaccine commercial vaccine at 24 days, G3: commercial vaccine at 24 days, and G4: commercial vaccine + oral vaccine at 24 days. On the first day, the piglets were weighed and, from the third day onwards, submitted to blood collections for the detection and quantification of anti-*Mycoplasma hyopneumoniae* IgG. Nasal swabs were collected to monitor IgA by ELISA, and oropharyngeal swabs were used to assess the bacterial load by qPCR. Biological samples were collected periodically from the third day of life until the 73rd day. At 41 days of life, 15 individuals of the same age, experimentally challenged with an inoculum containing *M. hyopneumoniae*, were co-housed with the animals from groups (1 to 4) in a single pen to increase the infection pressure during the nursery period. At 73 days, all piglets were euthanized, and lungs were evaluated by collecting samples for estimation of bacterial load by qPCR. Quantitative data obtained from physical parameters and laboratory investigation were analyzed by performing parametric or non-parametric statistical tests. Results indicate that animals from G2 showed smaller affected lung areas compared to G3. Animals from G2 and G4 had a low prevalence of animals shedding *M. hyopneumoniae* at 61 days of age. Additionally, no correlation was observed between lung lesions and *M. hyopneumoniae* load in lung and BALF samples in animals that received the oral vaccine, while a strong correlation was observed in other groups. In the present study, evidence points to the effectiveness of the oral vaccine developed for controlling *M. hyopneumoniae* in pig production under field conditions.

## 1. Introduction

*Mycoplasma hyopneumoniae* (*M. hyopneumoniae*) is the primary agent of porcine enzootic pneumonia (PES), which is one of the diseases of the respiratory complex of pigs [1]. PES mainly affects animals during the growing to finishing period that presents a chronic condition of non-productive cough, reduced weight gain, and feed conversion [2], as well as increased mortality and costs associated with the control and treatment of animals [3]. The bacterium acts on the mucous membranes of the respiratory epithelium by adhering to the cilia and changing the frequency of beats, promoting ciliostasis, epithelial cell death, modulating cellular and humoral immunity, and resulting in an exacerbated inflammatory reaction in the lungs [4].

Piglets are considered free of *M. hyopneumoniae* at birth, as in utero transmission has not been documented, and the first exposure events occur during the lactation period when piglets are in contact with sows that are shedding the microorganism, so this moment is considered of highest risk for the colonization of piglets by *M. hyopneumoniae*, related to the stress of parturition [5].

Currently, attempts to control PES involve monitoring the pathogen, attention to biosecurity, vaccination, and antimicrobial usage. However, control is difficult, considering that even applying all the measures, the prevalence of *M. hyopneumoniae* in swine production is high [4]. From an immunological point of view, the intramuscular vaccine helps to reduce the clinical signs; however, vaccines do not prevent the infection, only reduce the impact of the infection on the performance of the pigs [6]. The mechanism of action of commercial vaccines has been studied to understand the dynamics of the cellular and humoral immune response in these animals [7].

IgA is one of the defense mechanisms responsible for mitigating the adhesion of microorganisms to the epithelium and possibly one of the main components responsible for protecting the mucous membranes, whose production by commercial vaccines has been evaluated in recent studies. The role of mucosal immunization is to induce local production of IgA on the mucosal surface and on distant sites, made possible using the carrier-adjuvant [8].

Studies have been conducted to develop vaccines that act directly by stimulating an immune response that offers primary protection against the pathogen, involving the evaluation of the local IgA response [9]. Oral vaccines that promote induction of the respiratory and immune response have been developed and evaluated in several species [10,11,12], showing positive results. In pigs, the ability to stimulate the mucous membranes of different systems through oral immunization with other pathogens is reported. However, until now, there have been few studies focused on the development of oral vaccines against respiratory pathogens [13].

The Department of Applied Physics at the University of Sao Paulo—USP, in collaboration with the Butantan Institute, found that ordered mesoporous silica (SBA-15) is suitable for oral use and vaccine development. The chemical, mechanical, and thermal stability of silica, along with its high surface area and mesopores in the range of 8–50 nm with a narrow distribution, allows for the incorporation and release of molecules. This sets it apart from other adjuvants, such as aluminum hydroxide [14,15].

A specific oral vaccine against *M. hyopneumoniae,* using the adjuvant carrier SBA-15 has been developed, demonstrating its capacity to stimulate the immune system associated with the intestinal mucosa, showing promising results with the decrease in the extent of lung lesions at slaughter in animals experimentally challenged with a homologous strain (232) used in the vaccine composition [16].

This study aims to evaluate the use of the previously developed oral vaccine, experimentally under field conditions, against an uncharacterized field strain of *Mhyo*, considering the dynamics of the infection and the main challenges of commercial pig farming involving *M. hyopneumoniae.*

## 2. Result

Four pregnant females were challenged with a field strain of *M. hyopneumoniae* at 180 and 200 days of gestation. The litters born to the four *M. hyopneumoniae*-positive females were randomly assigned to groups as follows: G1 (gilt number 275), G2 (gilt number 350), G3 (gilt number 933), and G4 (gilt number 794). Throughout the study, the piglets were regularly weighed, and their physical parameters were evaluated. Periodic biological samples were collected from the third day of life until the 73rd day. All study groups received the intramuscular vaccine against *M. hyopneumoniae*. In addition to the commercial vaccine, G2 and G4 received the oral vaccine, while G1 received pure SBA-15 in an acidified PBS solution (pH 3.2). Group 3 was the only group that received only the intramuscular vaccine, as a standard vaccination protocol applied in commercial swine production. To increase infection pressure during the nursery period, 15 piglets of the same age as the others were challenged with *M. hyopneumoniae* on the 41st day of life and added to the pen with the animals from G1 to G4. Finally, at the conclusion of the study, on the 73rd day, all piglets were euthanized and underwent necropsy (Figure 1).

From the date the breeding females were transferred to the farrowing, the first signs of coughing were observed. Throughout the period, the four females whose litters composed the experimental groups presented sporadic episodes of dry cough. Female number 275, whose litter made up G1, presented recurrent episodes of anorexia from the 14th day after parturition.

The oral vaccine and SBA-15 + PBS were easily administrated, and there was no reflux or extravasation of any volume. Diarrhea did not occur at the time of vaccination or in the three days following the application of the compounds orally. On D24, when all animals received intramuscular vaccination, no temperature higher than 39.0 °C was recorded, and there were no noteworthy changes at the injection site up to 48 h after administration.

Information on the physical assessment of the piglets, changes in health condition, stool appearance, mucous membranes color, nasal discharge, and behavior are described in Table 1. Changes in health conditions were observed throughout the entire study, except for D51, in all groups. Diarrhea was observed from birth until the fifth day in G2 and G3, and from D27 to D41 in all groups, except in G3. Mucosal color change was observed only once on D51 in one of the G4 piglets. Nasal discharge was observed on D51 and D61 in all groups. Changes in behavior were observed on D25 and D27 in G1 and G4 and on D61 in G3 (Table 1).

Regarding body weight, no difference was observed between litters on the third day of life with mean in kilograms ± standard error of G1 (2.41 ± 0.1), G2 (2.32 ± 0.09), G3 (2.55 ± 0.07), G4 (2.54 ± 0.14) nor on D72 days among the groups G1 (11.27 ± 0.7), G2 (11.15 ± 0.71), G3 (11.94 ± 0.72), and G4 (11.57 ± 0.7) (*p* = 0.34); however, at 24 days, a difference in weight was observed between the vaccinated groups (*p* = 0.008) when G2 (6.06 ± 0.21) (mean ± standard error) did not differ from G1 (5.25 ± 0.29), G3 (6.32 ± 0.21), and G4 (6.48 ± 0.28), while the mean of G1 was smaller than G3 and G4. Regarding the daily weight gain (DWG) recorded during farrowing, from the 3rd to the 24th day, there was a lower weight gain in G1 (0.14 ± 0.01) when compared to G2 (0.19 ± 0.01), G3 (0.19 ± 0.01), and G4 (0.2 ± 0.01) which did not differ from each other by Tukey’s test (*p <* 0.001). The DWG throughout the nursery period indicated that there was no difference between groups according to Tukey’s test (*p* = 0.68).

Regarding rectal temperature, it was observed that on days 0, 5, 10, 41, 51, 61, and 71, no differences were recorded between the means of the groups by the Tukey test at the level of (*p <* 0.05) (Appendix A). On D4, the mean temperature of G3 was higher than that of G4. On D17, the mean temperature of G1 was lower when compared to the temperature of G4. Regarding D24, the temperature of G1 was lower than the temperature of the other groups. From D26 to D31, G1 averages were lower than G2 and G4, while G3 did not differ from the others.

Multiple comparisons between different rectal temperature measurement points within each group were performed to observe whether there was a variation in rectal temperature associated with the vaccination protocol. A decrease in rectal temperature over time was observed when comparing D0, D3, and D5 with the values recorded from D17 to D41 in G1, followed by an increase in temperature on D51 to D71 that differed from D41 to D61, which were superior to D24 and D31. In G2, the values recorded from D0 to D10 were higher than D17 to D71. In G3, the temperature of D0, D3, and D10 was higher than those from D17 to D41 and D71, and D5 was higher than D17 to D51 and D71. Additionally, in G3, an increase in rectal temperature was observed, with higher values on D51 compared to D41 and D61 when compared to D31 and D41. In G4, the values recorded from D0 to D10 were higher than D24 and from D26 to D51. Additionally, an increase in temperature was observed between D41 and D61 at the level of *p <* 0.05 by the generalized linear mixed model (Appendix A and Figure 1).

### 2.1. Anti-M. hyopneumoniae IgA and IgG in Nasal Swabs and Serum

Serum samples and nasal swabs collected from D3 to D72 were submitted to detection and relative quantification of immunoglobulin concentration (IgA and IgG), whose results were expressed in S/P value and compared between groups in each collection point and within each group over time.

Regarding the IgG quantifications in the piglets’ serum, differences were observed when comparing the medians between groups since the first collection. On D3, lower IgG concentrations were observed in G2, and the highest in G3, followed by G1 and G4, with all groups significantly differing from each other by the Kruskal–Wallis test at the level of *p <* 0.05. On D10, D17, and D24, both G1 and G3 were superior to G2 and G4. On D31, IgG titers were higher in G3 compared to the others, and G1 was higher than G2 and G4. On D41, all groups differed significantly from each other, with the highest concentration of IgG recorded for G3, followed by G1 and G4, with the lowest median observed for G2. On D51, it was observed that G3 had the highest measurement when compared to the other groups, followed by G1, which differed from G2 and G3. On D61, the S/P value in G3 was higher than in G2 and G4. Finally, on D71, no differences were found between the groups at the level of *p <* 0.05 by the non-parametric Kruskal–Wallis test (Appendix A).

Multiple comparisons over time demonstrated that for G1, the IgG levels on D71, D61, and D51 were significantly lower than the moments from D3 to D31. On D24 and D41, significant reductions in the titles were observed, compared to D3 and D17. In G2, the IgG levels on D71 were higher than those observed on D24 and D41. On D61, it was higher than on D41. Therefore, over time, in G2, there was a significant decrease in titles on D24 and D41 compared to moments from D3 to D17. In G3, a pattern similar to G1 was observed, with D71 and D61 significantly differing from D3 to D31 and D51 from D3, D17, and D31. On D41, a significant titer decrease was observed when compared to D3, D17, and D31 on G3. In G4, the levels of D71 differed only from D3, while D61 and D51 and D41 differed from D10 to D17 by Friedman’s non-parametric test for comparisons of repeated measures over time *p <* 0.05 (Figure 2 and Appendix A).

Regarding IgA, on D3, D41, and D51, no differences were recorded between the medians of the groups by the non-parametric Kruskal–Wallis test (*p* = 0.31). At D10, the S/P value was lower in G2 and G4 when compared to G1 and G3. On D17, the median of G4 was higher when compared to G1 and G2. On D31, G3 was higher than G4. On D61, the smallest measure was G3 when compared to the others. On D71, the median S/P was higher in G4 than in G3, while G1 and G2 did not differ from the others by the Kruskal–Wallis test at *p <* 0.05 (Appendix A).

Regarding multiple comparisons for IgA concentrations in the nasal swab, an increase in S/P values was observed in all groups, with the highest values presented on D71, D61, and D41. In G1, D71 and D61 differed from the values recorded from D3 to D41. In G2, the values on D71, D61, and D51 differed from D3 to D24. From D31 onwards, there was an increase in the S/P values, and, from that moment on, no significant differences were observed. As for G3, differences were observed when comparing D71 with moments from D3 to D24 and D61 and comparing D51 with D10 to D24. In G3, differences were also observed between D31 and moments from D10 to D24, demonstrating an increase in IgA levels from D31. In G4, differences were observed when comparing D71 with the moments from D10 to D31 and D61. On D51 (G4), significant differences were registered when compared to the moments from D10 to D24. Still, in G4, a significant increase was observed from D41 onwards, which differed from D10 and D24 by Friedman’s non-parametric test for comparisons of repeated measures over time at *p <* 0.05 (Appendix A and Figure 3).

### 2.2. Assessment of Lung Lesions

#### Determination of the Pneumonia Index

The lungs of the piglets in each group were evaluated individually, recording the degree of consolidation lesions in each lobe so that the total affected area could be calculated (Figure 4). Regarding the extension of macroscopic lung lesions (%), a difference was observed between G2 and G3. Significant differences were observed between G2 and G3 (*p* = 0.005), and a trend towards significance was noted between G4 and G3 (*p* = 0.07) (Table 2).

### 2.3. Estimation of Bacterial Load of M. hyopneumoniae in Oropharyngeal Swab, BALF, and Lung by qPCR

Samples of oropharyngeal swabs collected from piglets in groups 1 to 4 over the period from D3 to D72, as well as tracheobronchial lavage samples and lung fragments collected at euthanasia, were subjected to DNA extraction and detection of *M. hyopneumoniae* present in the samples (Figure 5).

Swab samples collected from D3 to D51 were negative for the presence of *Mhyo*, demonstrating that positive animals were not detected during this period. On D61, 20 days after the start of the challenge with animals experimentally exposed to lung homogenate, nine vaccinated piglets were detected as positive in qPCR. Altogether, two animals from G1, individuals number 5 and 10, totaling 13.33% (3.74 ± 37.88 % CI), one individual from G2, number 17, totaling 6.67% (1.19 ± 29.82 % CI), five positive individuals from G3, animals number 32, 33, 36, 40, and 41, totaling 35.71% of prevalence, and one individual from G4, number 54, resulting in a total of 13.33% (3.74 ± 37.88 % CI) (Table 3). The results of quantification of *M. hyopneumoniae* in the BALF, as well as in the lung, indicated that there was no difference between the vaccinated groups by Kruskal–Wallis test (Table 4).

### 2.4. Correlation Analysis

Correlation analyses were performed considering measurements observed over time (IgA, IgG, rectal temperature, and body weight) and also measurements obtained from post-mortem sample measurements and analyses. The correlation between body weight and the S/P value of IgA indicated that for the vaccinated groups (G1 to G4), the correlations were significant (*p* < 0.05) by the Spearman test. The value of rho varied from 0.30 for G3 to 0.58 for G1, indicating that the samples exhibit different levels of correlation between IgA and weight values. G1 has the strongest correlation, while G3 has the weakest correlation. For G2, rho was 0.43, and for G4, it was 0.41.

The correlation between IgA and IgG was analyzed in four groups (G1 to G4). Significant negative correlations were found in G1 (*p* < 0.001) and G3 (*p* < 0.01), with G1 having the strongest correlation (tau = −0.42) and G3 having the weakest (tau = −0.31). In G4, a significant negative correlation was observed (rho = −0.524, *p* = 0.045) for IgG and BALF quantification indicating that higher values of IgG would be correlated with lower concentrations of *M. hyopneumoniae* in BALF samples.

No correlations were found between bacterial loads in BALF and lung samples in any of the groups using Kendall’s test (*p* > 0.05). Although directly proportional correlations were observed between lung lesions at euthanasia and the quantification of *M. hyopneumoniae*, only in G1 and G3 (G1, *p*-value = 0.009 and R = 0.64; G3, *p*-value = 0.001 and R = 0.08), indicating a strong correlation between lung lesions and *M. hyopneumoniae* load. In G2 and G4, the recorded *p*-values were 0.43 and 0.51, respectively, by Pearson’s test. On the other hand, when analyzing the correlation between *M. hyopneumoniae* load in lung samples and lung lesions, no significant correlation was found in any of the groups.

## 3. Discussion

There is still little data in the literature regarding the use of SBA-15 associated with antigens in the composition of oral vaccines, especially for animals such as the one produced in the present study. A previous study of characterization and use of the same oral vaccine under laboratory conditions of controlled infection showed promising results in stimulating immunity and reducing lung lesions at slaughter [16].

Especially in pig production, zootechnical performance is a fundamental aspect that can suffer positive or negative impacts due to the action of drugs and vaccines [17]. Regarding the administration of the oral vaccine produced in the present study, no significant alterations in rectal temperature, no occurrence of diarrhea, and no alteration of the clinical picture were observed either in animals that received the oral vaccine at three days (G2) or in piglets vaccinated in the weaning group (G4). Changes in stool consistency were observed only on D31. However, they are likely associated with dietary changes after weaning [18]. Considering the post-challenge period in the nursery phase, in which experimentally challenged piglets were mixed with vaccinated piglets (G1-G4), the emergence of animals with nasal discharge at D51 and D61 in the four groups, animals that were positive for *M. hyopneumoniae* in samples collected after euthanasia, was observed, indicating a possible clinical picture of respiratory disease.

Regarding body weight and weight gain, differences were only observed in G1, which in turn may be associated with anorexia of female number 275, which may have resulted in lower quality and quantity of milk produced in some moments during the suckling period that reflected in the performance of the animals of the litter. Although there was no evidence of a difference between the weights of the piglets at slaughter, it is necessary to emphasize that the study only covered the period of farrowing and weaning, so, possibly, if the study was prolonged throughout growth and finishing, differences in weight gain and performance became more evident, which is a limitation of the present study.

As for body temperature, a reduction was observed over time in the groups, which is physiologically consistent with the variation in body temperature over the lifetime of pigs [19]. Variation in body temperature was observed between moments D41 and D51, being significant only in G1 and G3. Such a variation may be associated with an inflammatory process [20] since it is consistent with the initial challenge period caused by the mixing of infected animals. It is possible to observe, analyzing the plots and the recorded values, that such differences are present in the four groups, although there are indications that in G2 and G4, such temperature variation may have occurred more mildly when compared to the others. It was previously observed that the application of the oral vaccine acted in the modulation of the cellular immune response with an increase in the concentrations of anti-inflammatory cytokines, which was indicated as an effect possibly related to a lower incidence and extent of lung lesions in vaccinated animals [16].

In the swine species, passive immunity is of great relevance in the acquisition of antibodies that will help protect piglets during the farrowing period and the beginning of the nursery phase, with IgG being one of the main components involved in the process of transferring colostral immunity, therefore absorbed, shortly after birth [21]. In gilts, variability in passive immunity may result in differences between litters considering colostrum uptake [22]. In the present study, it was observed that more expressive amounts of anti-*M. hyopneumoniae* IgG were detected in G3 and G1 compared to G4, and especially in G2. This difference in the immunity of piglets from the different groups remained during practically the entire period, with a decrease in the concentration of immunoglobulins, which persisted even after vaccination of the groups, both with the oral vaccine and with the commercial intramuscular vaccine, with perhaps an only exception on the part of G2, in which an increase in the concentration of IgG was registered from D51, a fact that meant that on D71 no significant differences were registered between the groups regarding the concentration of such immunoglobulin.

Based on parallel coordinates plots and the analysis of the repeated measurements, it was observed that G2 was the only group that presented an increase in anti-*M. hyopneumoniae* IgG concentrations. In the other groups, there were no signs of an increase in antibody titers. However, it was not possible to determine that there was no stimulation of IgG production since humoral immunity may have raised antibody titers as the concentration of anti-*M. hyopneumoniae* IgG from colostral immunity decreased in G1, G3, and G4. Although IgG anti-*M. hyopneumoniae* does not have a direct role as a protective factor against *M. hyopneumoniae* infection, it is an indicator of cellular and humoral immune response stimulation. Considering passive immunity, a higher IgG titer may indicate a more significant contribution of colostrum-derived defense cells, which would affect piglet protection against infections [23]. 

Considering that IgG has no direct action in guaranteeing protection against *M. hyopneumoniae* and, therefore, mild effectiveness in controlling PES, much has been discussed about the effect of mucosal immunity in protection against colonization of the respiratory tract of pigs by controlling *M. hyopneumoniae* infection. In a previous study, an oral vaccine with SBA-15 produced following the same methodology as in the present study showed positive results regarding the stimulation of mucosal immunity in animals vaccinated at weaning against a homologous strain of *M. hyopneumoniae* (232) [16]. In the present study, the development of mucosal immunity in animals submitted, in addition to vaccination at weaning, to early vaccination on the third day of life, as well as the nonspecific effect of stimulation of IgA production by the carrier adjuvant (SBA- 15) were effects that have not been considered in the previously proposed model. Regarding the differences in the concentration of IgA between the groups, it was observed that, despite the differences up to D31, there was no increase in the concentration in any of the groups. From D31, increases in immunoglobulin concentration were observed, also in all groups, and significantly higher concentrations were recorded in groups G2 and G3 with more noticeable peaks, as observed in the parallel coordinate plots. From D41 onwards, there was an increase in IgA levels in the groups. Significant differences were observed from D61 with lower IgA concentration in G3, which had been vaccinated only with the commercial intramuscular vaccine. On D61 and D71, there were no differences between G1, G2, and G4, although, numerically, the values observed in G4 are higher than the others. Therefore, there are indications of a nonspecific stimulation by SBA-15 and that the oral vaccine produced and administered had a positive effect in stimulating the production of IgA when combined with the commercial intramuscular vaccine. Another point to be highlighted is that, numerically, the S/P values recorded in G4 are superior to the others. However, no significant difference was evidenced, probably due to the greater variability of the data in G4 and the number of animals per group, which are limitations for statistical models in animal studies.

A lower degree of lung injury, associated with the increase in IgA levels in animals that received the oral vaccine, was observed in animals that received the new vaccine in the previously conducted study [16]. In the present study, vaccinated piglets also showed a lower degree of lung injury, regardless of the vaccination protocol. In addition, both G2 and G4 animals had a smaller area of lung injury, indicating that oral vaccination, administered both early and at weaning, was effective in reducing lung lesions.

Correlation analyses were performed between S/P values for IgG and IgA, resulting in significant correlation only for groups G1 and G3. This suggests that in these groups, as maternal IgG levels decreased, an increase in IgA levels was observed. It is possible that in the other groups, where no significant IgG titers were derived from passive immunity, such a correlation was not observed.

The correlation between the *M. hyopneumoniae* load in the BALF samples and lung lesions in G1 and G2 indicates that infection by the bacterium is strongly associated with the occurrence of lesions in these individuals. In the groups that received the oral vaccine, G2 and G4, the correlation between *M. hyopneumoniae* and lung lesions, considering *Mhyo* quantification in lung and BALF samples, suggests that the observed lesions may not be directly related to *M. hyopneumoniae,* indicating the effectiveness of the oral vaccine in reducing lung lesions caused by *M. hyopneumoniae*.

By analyzing the detection of *M. hyopneumoniae* between the groups over time, no positive animals were observed until D41. Although field situations were simulated in the present study, complex events such as the spread of *M. hyopneumoniae* during farrowing may occur differently in a commercial production environment. The gilts in the present study were initially free from *M. hyopneumoniae* and were exposed to the pathogen only 20 days before parturition. Although positive during the farrowing period, they possibly shed less *M. hyopneumoniae* when compared to nulliparous females in the environment of production, in which the infection occurs endemically, resulting in early exposure to *M. hyopneumoniae*. Considering that the multiplication of *M. hyopneumoniae* has the characteristic of being fastidious [7] when infected early, the bacterial growth would occur for a more prolonged period, resulting in a higher bacterial load in the lung and, consequently, increasing the odds ratio that the bacteria is shed steadily and in greater concentration during the farrowing period. 

On the other hand, in the present study, there are indications that the spread of *M. hyopneumoniae* occurred from the challenge during the nursery period with the mixture of infected animals. After mixing on D41, clinical respiratory signs were observed, and the first positive piglets were identified on D61 in oropharyngeal swab samples. Numerically, a difference can be observed when comparing the proportions of positive animals between groups G3, G2, and G4, although based on the confidence interval, it is not possible to show a significant difference. A limiting factor for comparisons between proportions is the number of samples analyzed, and, in the case of a study with animals, there are few options for analyzes and models that are actually appropriate to determine differences between groups with a reduced number of experimental units.

Based on the present study, measures based on vaccination were essential in protecting the animals, although they were not fully effective in inhibiting the dissemination or infection by *M. hyopneumoniae*, demonstrating that the control of the bacteria depends, in addition to vaccination, on other joint actions such as the rational use of efficient antimicrobial molecules [24] and especially effective biosecurity measures in production [6]. Given the results of the present study, there is evidence that oral vaccination with a vaccine produced using the carrier-adjuvant has been effective in reducing lung lesions and the dissemination of *M. hyopneumoniae* in production. Future studies may explore formulations based on SBA-15 in other ways and associated with *M. hyopneumoniae* antigens produced from national strains facing homologous and heterologous challenges. 

## 4. Material and Methods

### 4.1. Animal Selection

For the study, six pregnant primiparous females with 80 days of gestation were acquired from a producer located in the municipality of Itu, SP, and sent to the Swine Medicine Laboratory, where they remained in gestation pens (0.3 female/m^2^) for up to 10 days before the expected parturition date. After this period, the gilts were transferred into individual farrowing pens. Upon arrival, they were subjected to blood samples collection to obtain serum for antibody quantification and laryngeal swab collection to verify the females’ status regarding *M. hyopneumoniae* infection. After farrowing, the groups were defined, and the litters were kept with the gilts during the farrowing phase (24 days). After weaning, the females returned to the collective pens, and the four experimental groups (G1 to G4) were kept together in a single nursery pen until they were 72 days old (2 piglets/m^2^). At 41 days, 12 animals, positive for *M. hyopneumoniae*, randomly chosen from the litters of the other two breeding gilts, were mixed with the other piglets from G1 to G4. Strict biosecurity measures were followed to avoid cross-infection and minimize external interference, such as the use of specific, clean clothes and not having contact with any other pigs during the experimental period. The animals received feed according to the production phase, free of antibiotics, and water ad libitum. All procedures of this study were approved by the Ethics Committee for the Use of Animals (CEUA), Faculty of Agrarian and Veterinary Sciences, Unesp Campus of Jaboticabal–SP, under protocol number 00915/20. 

### 4.2. Experimental Design

Pregnant females were, at 94 and 100 days of gestation, exposed to *M. hyopneumoniae* via the intratracheal and nasal route using a positive inoculum (10^5^ copies∕μL) for *M. hyopneumoniae*, produced from lungs with lesions suggestive of PES, obtained from a slaughterhouse located in the city of Guariba, SP. A total of 60 piglets born to four females positive for *M. hyopneumoniae* were divided into four groups (n = 15) and submitted to different vaccination protocols as follows, G1: SBA-15 in acidified PBS (pH 3.2) without *M. hyopneumoniae* antigens + commercial vaccine at 24 days, G2: oral vaccine at the third day + commercial vaccine at 24 days, G3: commercial vaccine at 24 days, and G4: commercial vaccine + oral vaccine at 24 days. On the first day, the piglets were weighed, and from the third day on, they were subjected to blood collections for the detection and quantification of anti-Mycoplasma hyopneumoniae IgG, as well as nasal swabs for monitoring IgA by ELISA and oropharyngeal swabs to estimate the bacterial load of *M. hyopneumoniae* by qPCR. Biological samples were collected periodically until the 72nd day. At 41 days of life, 12 individuals of the same age from two other pregnant females previously exposed to *M. hyopneumoniae* were mixed with the animals from groups (1 to 4) to increase the infection pressure during the nursery period. At the end of the study, at 72 days, all piglets were euthanized and necropsied for evaluation of the lungs and collection of lung samples for estimation of bacterial load by qPCR. Samples of bronchoalveolar fluid (BALF) were also collected and submitted to the qPCR test.

### 4.3. Inoculum

For the inoculum production, five lungs of animals at approximately 150 days of age, which presented consolidation lesions in varying degrees in the portions of the right and left cardiac lobes, were collected in a slaughterhouse placed in the municipality of Guariba, SP. The lungs were sent refrigerated to the Swine Medicine Laboratory. Fragments of the apical and cardiac lobes of the lungs were selected, with the aid of a sterilized scalpel blade, sectioned into smaller portions of approximately 1 mm^3^, which in turn were transferred to a clean and sterilized plastic package, to which 500 mL of PBS (pH 7.2) heated to 37 °C were added, homogenizing the content, pressing the fragments and stirring the volume. The homogenate was then filtered using a clean and disinfected sieve, and part of the filtered volume was aliquoted in fractions of 1 mL in plastic tubes free of DNAse and RNAse intended for analysis to estimate the concentration of *M. hyopneumoniae* in the inoculum. DNA from the inoculum samples was immediately extracted [25] and subjected to qPCR, indicating a quantification of 10^5^ copies/µL. 

### 4.4. Challenge of Pregnant Females

Before the challenge, the pregnant females were subjected to blood sample collection to obtain blood serum and oropharyngeal swabs. Blood samples were collected using a vacuum tube with clot-activating gel through a jugular vein puncture using an attached 25 × 0.8 mm needle. The oropharyngeal swab samples were collected using a mouth opener and a laryngoscope by rubbing the cotton swab in the region posterior to the epiglottis of the females.

The pregnant females were challenged on two occasions because they had been vaccinated on the farm of origin at 180 and 200 days of life with the vaccine RespiSure (Pfizer, New York, NY, USA), which was also verified by testing positive in the ELISA with S/P values for anti-*M. hyopneumoniae* IgG in: 1.9 × 10^1^ (female 275), 6.6 × 10^0^ (female 350), 5.7 × 10^0^ (female 933), and 2.2 × 10^0^ (female 794). Although vaccination does not prevent infection, it was expected that not all females would become infected. The litters born to the four *M. hyopneumoniae*-positive females were randomly assigned to groups as follows: G1 (gilt number 275), G2 (gilt number 350), G3 (gilt number 933), and G4 (gilt number 794). Additionally, the litter of two negative females for *M. hyopneumoniae* composed a group of animals experimentally challenged against *M. hyopneumoniae.*

The first exposure of the females was performed about 20 days before the expected parturition (94 days of gestation) and the second 10 days before farrowing (104 days). At both times, the gilts were exposed to the lung homogenate through the nasal and tracheal routes. For that, the females were restrained using a rope and with the oral cavity opened using a mouth opener and a laryngoscope. In each exposure, 10 mL of inoculum were then administered directly into the trachea of the females using an adapted insemination pipette. After exposure through the tracheal route, nasal administration was performed with 5 mL in each nostril, instilling the volume gradually and keeping the female with her head elevated so that there was no extravasation of the administered volume.

About 24 h before farrowing, the females were submitted to oropharyngeal swab collection to determine their status for *M. hyopneumoniae* infection and define the groups. Swabs were collected and transferred to 2 mL plastic tubes containing PBS (pH 7.2) and then immediately processed for DNA extraction and subjected to qPCR for *M. hyopneumoniae* detection and quantification in the sample.

qPCR results indicated that of the six females challenged, four of them were positive 24 h before parturition. The four females gave birth on the same date. A total of 15 piglets were born in each of the litters from sows: 275, 350, and 933, while sow 794′s litter consisted of 16 piglets. Litter standardization was not carried out, nor were transfers of piglets between sows.

Before the collection of samples and the definition of the formulations that each group would receive, all animals, from G1 to G4, were weighed, their temperature was measured, nasal swabs and blood were collected, and they were identified with earrings, defining a total of 15 piglets per group. The drawing of formulations among the different litters was carried out in the R Project software version 4.0.3 using the “sample” function. The vaccination protocols were randomly defined to each group as follows: G1: oral SBA15 + commercial vaccine at 24 days, G2: oral vaccine on the third day + commercial vaccine at 24 days, G3: commercial vaccine at 24 days, and G4: commercial vaccine + oral vaccine at 24 days.

### 4.5. Production of the Oral Vaccine

The oral vaccine production process followed the same protocols described by Mechler-Dreibi [16]. A pathogenic strain of *Mycoplasma hyopneumoniae* (232) was purchased from Iowa State University and certified free of other pathogens. Initially, a pre-culture was performed using 5 mL of Friis medium. Then, four flasks containing 100 mL were inoculated, with 1 mL of the previous culture kept at 37 °C in an oven. The procedure was performed twice to obtain all the necessary culture volume. The color was observed daily, which showed changes about seven days after inoculation. The flasks were removed from the oven when the culture medium turned orange. A volume of 700 mL of Friis medium with *M. hyopneumoniae* was obtained. The final volume was transferred to 50 mL falcon tubes and kept at −80 °C until processing. A sterility test was conducted on blood agar and McConkey media kept in the incubator at 37 °C for three days to evaluate the microbiological growth indicative of contamination. *M. hyopneumoniae* was also cultivated in a modified Friis Agar solid medium, and in seven days, it was possible to observe the presence of bacterial colonies.

The concentration of *M. hyopneumoniae* was determined through successive dilutions of the samples in Friis medium, varying from 10^−1^ to 10^−7^. In addition to estimating the quantification of *M. hyopneumoniae* through successive dilutions, samples were submitted to qPCR. The results obtained from qPCR indicated concentrations between 1.8 × 10^6^ and 1.3 × 10^7^ copies/µL. The culture was centrifuged to form a bacterial cell pellet at 13,700× *g* for 45 min at 4 °C. Three washing steps with PBS (pH 7.4) were performed, and the *M. hyopneumoniae* was suspended in PBS. The suspension was subjected to a sonication process (Soni -tech Ultrasonic Cleaning) at a frequency of 20 Hz for 1 min, with one-minute intervals between processes. To determine the protein concentration in the bacterial lysate, the Bradford method was applied (Thermo Fisher Scientific, Rockford, IL, USA) followed by spectrophotometer reading (NanoDrop One, Thermofisher Scientific, USA), which provided a concentration of 1125 µg/ mL.

About seven days before the oral vaccination administration, the dry fraction of the vaccine was prepared, measuring the number of proteins suspended in PBS (pH 7.2). The dose used for the production of the oral vaccine was 200 µg per piglet, mixed with silica in the proportion 1:35 defined in previous studies [14,26]. *Mycoplasma hyopneumoniae* protein lysate was added to SBA-15, previously macerated and activated, and a drying step at 37 °C for 72 h was performed to obtain the dry fraction [16].

About two days prior to administration, the complete oral vaccine and SBA-15 compound in acidified PBS without *M. hyopneumoniae* antigens were prepared. For the production of the complete oral vaccine, the Eudragit^®^ L30 D-55 polymer (Evonik) was added to the dry fraction consisting of SBA-15 + *M. hyopneumoniae* lysate, and the mixture was macerated until a paste was formed. After the addition of Eudragit^®^, a new drying step was carried out at 37 °C for 48 h until the content was completely dried. The dry antigen + silica fraction, coated with Eudragit^®^, was macerated again, and acidified PBS (pH 3.2) was added with 2M HCl. The addition of the acidifier was performed just moments before product administration to the animals. Acidified PBS (pH 3.2) was added, considering a final vaccine volume of 3 mL per individual. Before administration, the vaccine was vigorously shaken to ensure complete homogenization of the mixture in the bottle.

For the production of the compound without *M. hyopneumoniae* antigens, the silica was weighed on an analytical balance considering the same proportion of silica administered to animals that received the complete oral vaccine (7 mg per animal). Pure SBA-15 was macerated using a glass rod, and acidified PBS solution (pH 3.2) was added. In both the production of the complete oral vaccine and the pure SBA-15, the addition of the acidifier was performed just moments before administration to the animals. In both cases, acidified PBS (pH 3.2) was added, considering a final solution volume of 3 mL per individual. Before administration, the bottles containing the complete vaccine and the acidified SBA-15 without antigens were vigorously shaken for complete homogenization of the mixtures in the bottles.

### 4.6. Vaccination of Piglets

The administration of the oral vaccine and SBA-15 + acidified PBS was performed using a 5 mL syringe coupled to a urinary catheter 16 Fr cut with the aid of disinfected scissors. The probe size was reduced to three centimeters, enough to direct the content directly into the oral cavity of the animals, similar to the administration using a pump bottle dispenser. 

The commercial vaccine used was Hyogen^®^ (CEVA Santé Animale, Libourne Cedex, France), a mineral oil emulsion with non-toxic LPS from Escherichia coli J5 as an immunostimulant and inactivated *M. hyopneumoniae* BA 2940-99 field isolate as an antigen.

The commercial Hyogen^®^ vaccine (batch: 005/21, manufacture: 4/21, and expiry date: 4/23) was applied following the manufacturer’s guidelines, with 2 mL per individual being administered to all piglets in groups G1 to G4 by intramuscular route in the neck region using 25 × 0.8 mm disposable needles and 5 mL syringes. In G1 and G4, the application of the commercial vaccine was performed shortly after the application of the oral vaccine.

The vaccine injection site was evaluated beforehand and after 48 h. Regarding the volume administered intramuscularly, no extravasation of the vaccine was observed in any of the animals. After the oral vaccine and SBA-15 + acidified PBS application, the individuals remained under observation for about 1 h to observe any occurrence indicative of reflux or vomiting in these animals.

For three days, the piglets were evaluated for signs of diarrhea and submitted to body temperature measuring 48 h after the application of the commercial vaccine.

### 4.7. Assessments and Sample Collections in Piglets

Nasal swabs and blood samples were collected over time. Then, at D72, piglets were euthanized and necropsied. The animals were weighed at 3 days of life, 24 days, and 71 days. At three days old, the piglets received 2 mL of iron dextran solution (Ferdex, Fabiani Saúde Animal, São Paulo -SP) and Baycox ^®^ 5% (Elanco, Brazil). Clinical assessments were carried out periodically and consisted of temperature measurement, evaluation of the general state of the animals (classified as good, regular, or bad), mucous membrane color, evaluation of the animals’ behavior (normal, apathetic, or uncoordinated), evaluation of the occurrence of diarrhea (normal, mild diarrhea or severe diarrhea), and the presence of nasal or ocular discharge.

#### 4.7.1. Blood Collection

At each time point, after the physical evaluation of the animals, they were submitted to the collection of blood samples by puncturing the jugular vein with sterile needles, size 25 × 0.8 mm, and deposited in vacuum tubes (red cap), free of anticoagulant and with clot activating factor and centrifuged at 3000 rpm for 10 min (Centrifuge 5804 R, Eppendorf ^®^, Hamburg, Germany). Blood serum was aliquoted in 1 mL fractions into 2 mL sterile microtubes in duplicate and stored in a −20 °C freezer and later analyzed for the detection and quantification of anti-*M. hyopneumoniae* IgG using the m.hyo Ab Test Kit (IDEXX Laboratories, Inc., Westbrook, ME, USA, Lot: DT583; Expiration: 23 September 2022).

#### 4.7.2. Nasal and Oropharyngeal Swab Collection

Nasal swab collections were performed using sterile cotton swabs. With the individuals properly restrained, the swab was lightly rubbed on the nasal mucosa of both nostrils. At the time of collection, contact with other external areas of the animal’s nostril was avoided. Immediately after collection, the swabs were placed in sterile microtubes with a buffer solution (PBS—pH 7.2) and transferred to a freezer at −20 °C, where they were kept until the moment of analysis for IgA Ab detection using the indirect ELISA technique using the commercial M.hyo kit Ab. Test (IDEXX Laboratories, Inc., USA, Lot: DT583; Expiration: 23 September 2022) standardized [16].

Swabs were collected from the animals to confirm the presence or absence of *Mycoplasma hyopneumoniae* through the detection of the p102 gene. For collection, sterilized and individually wrapped plastic swabs with a cotton tip were used. With the animal physically restrained and with the aid of a laryngoscope with a light source, the oral cavity was opened. The lingual torus was pressed downwards to facilitate access to the oropharynx region, very close to the tonsils, and then the swabs were lightly rubbed in rotating movements on the swab stem so that the entire extremity had contact with the mucosa. Contact with other internal areas of the animal’s mouth was avoided when introducing and removing the swab, and immediately after it was placed in a sterile microtube, duly identified with the animal’s number and time of collection, the swab stem was cut with the aid of clean scissors with 70° alcohol to each animal and stored at −20 °C. Oropharyngeal swab samples were subjected to DNA extraction by qPCR for the detection and quantification of *M. hyopneumoniae DNA.*

### 4.8. Nursery Phase and Sanitary Challenge

The piglets were weaned at 24 days of age, after which the sows were transferred, and the piglets from G1 to G4 were housed together in a pen. Two out of the six female litters that were not initially selected to join the study groups were separated at 24 days of age and placed in a separate nursery. About seven days after vaccinating groups 1 to 4 (at 31 days of age), 15 piglets from the two additional litters that did not belong to any study group were separated from the 60 vaccinated pigs and challenged with 10 mL of the same inoculum at a concentration of 10^5^ copies/µL, administered intratracheally to pregnant sows. The challenged piglets were housed separately from those in G1 to G4.

Ten days after the challenge, at 41 dpi, the challenged animals (n = 15) were co-housed with the vaccinated animals, G1 to G4, in a single nursery pen, totaling 75 animals. The aim of placing infected animals together with those subjected to vaccination protocols was to simulate the natural transmission of the pathogen between piglets that were exposed experimentally to *Mhyo* (G1 to G4) and those that were not exposed through the air and nose-to-nose contact.

### 4.9. Euthanasia of Piglets, Assessment of Lungs, and Collection of Samples

At the end of the experimental period, at 71 days of age, all animals were euthanized. For euthanasia, the piglets received an intramuscular injection of acepromazine (5 mg/kg), intramuscular administration of xylazine and ketamine (5 mg/kg and 7 mg/kg, respectively), and intravenous administration of propofol (5 mg/kg). Finally, 2% lidocaine was injected intrathecally to promote cardiorespiratory arrest. The animals were then subjected to bleeding and necropsied to obtain the respiratory set (trachea + lung), which was subjected to bronchoalveolar fluid (BALF) collection. BALF collection was performed by introducing 20 mL of 1X PBS into the cranial portion of the trachea. After pouring all the liquid, the lung was lightly massaged, and the liquid was aspirated by pipette, recovering an approximate volume of 10 mL. The aspirate was transferred to 15 mL Falcon tubes, stored in a freezer at −80 °C, and aliquoted in duplicate in 2 mL graduated microtubes, free of dNases and rNases (Kasvi, Brazil), for qPCR analysis.

The lungs were evaluated for the presence and extent of lesions by a unique blinded investigator. The extent of lung lesions was quantified using the European Pharmacopoeia method, in which the percentage of each affected area of the lobe was multiplied by lobe relative weight and summed to provide the percentage of the total weight of the affected lung [27]. Lung tissue fragments for qPCR were obtained with individual scalpel blades and sterile tweezers. The tweezers were kept in boiling water between collections and previously disinfected using 70% alcohol. These fragments were collected in duplicate and transferred to dNase and RNAse-free plastic packages kept at −80 °C. Subsequently, the lung fragments were processed to standardize the weight and region from where the lung fragments were obtained and destined for extraction and qPCR.

### 4.10. Detection and Quantification of IgA and IgG Antibodies by ELISA

Detection and quantification analyses of antibodies (IgA and IgG) were performed in the nasal swab and blood serum, respectively. To detect serum IgG, the commercial kit M.hyo Ab test (IDEXX Laboratories, Inc., Westbrook, ME, USA) was used. For the detection of IgA Ab in nasal swabs, an adaptation using the same commercial IgG detection kit was used as described previously [15]. For IgA detection, initially, the plates were blocked with 1.5% ovalbumin in PBS, followed by incubation at 37 °C for 30 min. Negative and positive samples of nasal swabs from a previous experimental infection study were selected and homogenized in the form of a pool to compose a negative (CN) and a positive (CP) control that was aliquoted to be used in all ELISA plates. To detect IgA Ab in nasal swabs, 100 μL of the sample’s liquid fraction were used, as the swabs were deposited in 500 μL of PBS, which were quickly homogenized in a vortex and placed, without further dilution, in each microplate well. The conjugate from the kit was replaced by an immunoenzymatic conjugate of goat anti-Pig IgA Antibody HRP Conjugated (Bethyl Laboratories Inc., USA) at a dilution of 1:500 using the diluent provided by the kit. At the end of the respective protocols, both the IgG and IgA analysis plates were read in a iMark microplate reader (Bio-Rad Laboratories Inc., Hercules, CA, USA) at a wavelength of 650 nm.

The mean optical densities (OD) for each of the test samples were related to the OD found for the negative and positive controls (NC$\underline{x}$, PC$\underline{x}$) to calculate the S/P values (positive sample ratio) according to the formula: S/P = oDs − NC$\underline{x}$/PC$\underline{x}$ − NC$\underline{x}$. The threshold between positive and negative samples was calculated from the value of S/P NC$\underline{x}$ + 2 × standard deviation. Serum samples and nasal swabs were considered positive if S/P > 0.3.

### 4.11. DNA Extraction from Nasal Swabs, Lung Samples, and BALF

DNA extraction was performed using the in-house Tris–- HCl protocol [25]. For nasal swabs and BALF samples, centrifugation (Centrifuge 5804 R, Eppendorf, Germany) at 13,000× *g* at 4 °C for 20 min was performed before the DNA extraction protocol. For lung samples, 0.05 g of tissue were used. Extracted DNA was stored at −20 °C until qPCR analysis. The concentration of DNA present in the extraction product was evaluated utilizing spectrophotometry, with the aid of the Thermo Scientific NanoDrop 2000 (Thermo Fisher Scientific^®^, Waltham, MA, USA), with the exclusion factor being samples that did not reach a purity of 1.8 to 2.0 in the 260/280 ratio before performing the qPCR. To rule out the presence of inhibitors in the extracted DNA samples and the occurrence of false negatives in the qPCR for *M. hyopneumoniae*, all samples were subjected to a conventional PCR targeting the endogenous gene Glyceraldehyde-3-phosphate dehydrogenase (*gapdh*) [28]. Amplified products of the 437 bp *gapdh* gene were detected after horizontal electrophoresis on a 1% agarose gel stained with ethidium bromide (0.5 μL/mL) in TEB running buffer pH 8.0 at a current of 90 V/50 mA for 90 min.

### 4.12. qPCR Analysis

Absolute real-time quantitative PCR analysis (qPCR) was used to detect *Mhyo* in oropharyngeal swab samples, lung fragments, and BALF. For *M. hyopneumoniae*, the primers used in the reaction were designed to the bacterium p102 adhesion protein gene sequence. All samples were tested in duplicate, and the qPCR analysis followed a previously optimized published protocol [29] adapted [30]. The nucleotide sequences used were forward primer 5″-AAGGGTCAAAGTCAAAGTC-3″, reverse primer 5″-AAATTAAAAAGTGTTCAAATGC-3″, and hydrolysis probe 5″-FAM-AACCAGTTTCCACTTCATCGCC-§BHQ2-3″ [29].

Only results in which the standard deviation between duplicates was less than or equal to 0.5 were accepted. Otherwise, samples were retested in triplicate. Sterile ultrapure water (Nuclease-Free Water, Promega^®^, Madison, WI, USA) q.s.p. was used as a negative control in the qPCR reactions. Serial dilutions were performed to determine the standard curve generated with different concentrations of synthetic DNA (gBlock^®^, IDT, Iowa City, IA, USA) containing the target sequence (10^7^ copies/μL to 10^1^ copies/μL), which were also used as positive controls. Synthetic DNA was diluted as per the manufacturer’s recommendations and maintained at a stock concentration of 10^7^ molecules/μL.

Quantification was performed using 10-fold serial dilutions (starting at 10^7^ to 10^1^ copies∕μL) of synthetic DNA (gBlock ^®^, IDT, Iowa City, IA, USA) containing the 150 bp fragment amplified by the pair of primers used in qPCR. The quantification data based on the standard curve were only validated if the reaction efficiency was between 90 and 105% [31].

## 5. Statistical Analysis

The variables in each group for each time point were assessed for normality and homoscedasticity by the Shapiro–Wilk and Bartlett tests, respectively. The difference among means was calculated by Tukey’s test (*p* < 0.05). Variables that did not meet the assumptions were submitted to the non-parametric Kruskal–Wallis test (*p* < 0.05), and in cases where significance was observed, the Dunn test (Post hoc) was applied. The Friedman non-parametric test [32] and generalized linear mixed models (GLMMs) were used for multiple comparisons among the time points *p* < 0.05. For proportion analysis, differences between groups by date were calculated using Wilson’s scoring interval method. Comparison between two paired measurements was performed using the Wilcoxon paired test. Values of measurements recorded over time and at euthanasia were subjected to a correlation analysis using the parametric Pearson and non-parametric Spearman tests, as well as Kendall analyses, to measure the association between the two variables. For this, the R software version 4.0.3 was used with the following packages: “agricolae” [33], “LmerTest” [34]; “arm” [35]; “Emmeans” [36]; “Car” [37]; “Nortest” [38]; “MASS” [39] with R software version 4.0.3.

## 6. Conclusions

The present study showed the positive effects of the administration of the oral vaccine, produced with SBA-15 carrier-adjuvant and strain 232 antigens associated with a standard intramuscular vaccination protocol, against a natural challenge with a field strain of *M. hyopneumoniae*. Higher levels of mucosal immunity and reduction of lung lesions at euthanasia were observed, being more evident results in animals that received the oral vaccine early. The application of the oral vaccine under field conditions can, even associated with the conventional vaccine protocol, provide better control of *M. hyopneumoniae* infection, substantially reducing the effects of infection in swine and possibly promoting better performance and quality of life results for pigs in commercial production environments.

## Data Availability

Not applicable.

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
