# Peer review of "Use of Nanostructured Silica SBA-15 as an Oral Vaccine Adjuvant to Control Mycoplasma hyopneumoniae in Swine Production"

_ijms, 2023, doi:10.3390/ijms24076591_

Round 1

Reviewer 1 Report

Ijms-2170508-peer review

The manuscript by Storino et al. titled “Use of nanostructured silica SBA-15 as an oral vaccine adjuvant to control Mycoplasma hyopneumoniae in swine production” described vaccine efficacy of novel oral vaccine (reference 15) in combination with the commercial vaccine. The authors presented the efficacy of the oral vaccine by prophylactic immunization protocol in reference 15.  In this manuscript, the authors compared efficacy of prophylactic/therapeutic immunization protocols. The authors used a complex protocol to mimic the field condition and assayed physical and bacteriologic parameters.  I believe the information presented here will be helpful in the agriculture and veterinary science field. The manuscript is written well. However, additional data or revision of the figures will need to improve the quality of the manuscript.   

1)    Did authors investigate the correlation of antibody levels and each clinical severity score such as BALF bacterial copies (by histology scores, p102 gene PCR score, body temperatures, body weights, or individual pneumonia index)? The information would be useful for searching the predictable biomarkers. 

2)    Oral vaccination in G3 (D3) or G4 (D24) show better protection in IgA and lung histology. However, IgG levels are lower than G1 and G3.  Also, IgG and IgA seem to correlate inversely. 

a.     Did the authors try to evaluate the correlation of antibody titles (IgG and IgA) at each time point? Is IgA better predictive biomarker?

b.     Are higher IgG levels in the early phase due to maternal transfer?

c.     Which protocol, days 3 or 24, providing the better protection?

d.     Please consult with the professional statistician. They may suggest different statistical approaches according to the data distribution.  

3)    Please explain why the authors did not include “no vaccination control” in the study. Also, was the study performed only once or repeated? Please add this reproducibility information and rationale if performed once.

4)    Because of the manuscript's structure, the protocol scheme (Scheme 1) is presented after the result. 

a.     I recommend moving scheme 1 to the first section of the results to help the readers understand the protocol. 

b.     In scheme 1, please add the legends of each symbol, such as oral immunization, commercial vaccine immunization, and blood draw.

c.     The dates in the scheme 1 are not chronological order, D3, and next D1 and again D1, The day 3 G2 received oral vaccine

d.     In the abstract and method, the authors indicate that G1 received silica particles without antigen on day 3. However, the scheme indicated day 2 in the scheme 1.

e.     In G4, there is no indication of day 24 oral vaccine in the scheme 1.

f.      There is no sign of commercial vaccine administration on day 24 in the scheme 1.

5)    Please add a brief description of the oral vaccine and SBA-15 in the introduction

6)    Table 1. The subtitle should be “classification.”

7)    Tables 2, 3, and 4.  The footnote “Means followed by the same letter……(p<0.05)” is not clear. The meaning of superscripts “a” and “b” is also not explained and hard to understand the results. Please reconsider to simplify the statistical comparison.

8)    Figures 1, 2, 3, 4, and 5. The figure legends are missing.  Please describe a brief protocol and Statistical analysis if performed.  Also, explain the abbreviation, e.g., Inoc in Figure 4 (Inoc=CHL?) and PUL and LTB in Figure 5. 

9)    Figure 2 IgA and Figure 3 IgG seem to be switched. X-axis labels are missing in both figures.

10) Figure 4.5., Please indicate each small dots represented in the figure legends.

Author Response

The manuscript by Storino et al. titled “Use of nanostructured silica SBA-15 as an oral vaccine adjuvant to control Mycoplasma hyopneumoniae in swine production” described vaccine efficacy of novel oral vaccine (reference 15) in combination with the commercial vaccine. The authors presented the efficacy of the oral vaccine by prophylactic immunization protocol in reference 15.  In this manuscript, the authors compared efficacy of prophylactic/therapeutic immunization protocols. The authors used a complex protocol to mimic the field condition and assayed physical and bacteriologic parameters.  I believe the information presented here will be helpful in the agriculture and veterinary science field. The manuscript is written well. However, additional data or revision of the figures will need to improve the quality of the manuscript.   

We appreciate your feedback and we thank you in advance for your suggestions, which have been quite relevant and have helped us improve our work.

1)    Did authors investigate the correlation of antibody levels and each clinical severity score such as BALF bacterial copies (by histology scores, p102 gene PCR score, body temperatures, body weights, or individual pneumonia index)? The information would be useful for searching the predictable biomarkers. 

Thank you for your suggestion. Correlation analyses were conducted between the parameters recorded in the present study and have been included in the results section (lines 335-359). Based on the observed results, changes have been made to the discussion section (lines 458-468) and the abstract of the paper (lines 34-39).

2)    Oral vaccination in G3 (D3) or G4 (D24) show better protection in IgA and lung histology. However, IgG levels are lower than G1 and G3.  Also, IgG and IgA seem to correlate inversely. 

  1. Did the authors try to evaluate the correlation of antibody titles (IgG and IgA) at each time point? Is IgA better predictive biomarker?

Correlation analysis was conducted and an inverse correlation was indeed observed. Contrary to expectations, a substantial increase in IgG was not observed during the study period. What was observed was a reduction in colostral transfer antibody levels, especially in groups 1 and 3. Therefore, the inversely proportional correlation only in the aforementioned groups may be directly related to the fact that the vaccination protocols induced some immune response with an increase in IgA titers, while colostral IgG naturally declined over time. Considering that one of the main ways to protect against Mhyo adhesion and infection is through mucosal immunity. In this regard, IgA detection can be an important biomarker in the study of oral vaccine, as it is one of the main elements of mucosal immunity and potentially a key factor in controlling M. hyopneumoniae, which is not fully addressed by many commercial vaccines known to have limitations in their efficacy in controlling the pathogen.

  1. Are higher IgG levels in the early phase due to maternal transfer?

We believe so. Although we attempted to standardize the M. hyopneumoniae-vaccinated breeding females, as described in the literature, there is variability regarding the transfer of passive immunity, which we believe may have occurred in the present study.

  1. Which protocol, days 3 or 24, providing the better protection?

Based on the data obtained from descriptive and correlation analyses, we conclude that both the protocols applied to groups 2 and 4 were effective. At field, early protection against M. hyopneumoniae would be highly advantageous, particularly during the parturition period which is considered high-risk for transmission. In the present study, although not statistically significant, a smaller number of positive animals was detected in G2 on D61, and the extent of lung lesions was also reduced compared to G3. Despite our attempt to simulate a high infection pressure environment during parturition, it is possible that vaccinated females were able to resist the infection better under the actual field conditions, leading to a lower excretion of M. hyopneumoniae in the environment. Therefore, the true effect of early oral vaccination, though significant, may have been underestimated in the present study. Consequently, we believe that the protocol applied to group 2 may yield even more effective results under field conditions.

  1. Please consult with the professional statistician. They may suggest different statistical approaches according to the data distribution. 

Thank you for the suggestion. Changes have been made to the data presentation, such as new graphics and correlation analyses were performed to better understand the results obtained.

3)    Please explain why the authors did not include “no vaccination control” in the study. Also, was the study performed only once or repeated? Please add this reproducibility information and rationale if performed once.

Thank you for the comment. Regarding the non-use of an unvaccinated control group, we justify it by some factors. Not only one, but two groups with animals vaccinated only with oral vaccines, at 3 and 24 days, would need to be included, increasing the complexity of the study and introducing new variables, taking into account that the field strain of M. hyopneumoniae was unknown. Thus, we believe that the methodology would deviate from the study objective that was evaluate the use of oral vaccine as a complement to commercial vaccination therefore, in line with reality of commercial swine production, where commercial vaccines are massively used. Finally, the study conducted by Mechler-Dreibi et al, 2021 had already analyzed the effect of using only oral vaccine in animals challenged with M. hyopneumoniae. Regarding the second question, the study in question was performed only once.

4)    Because of the manuscript's structure, the protocol scheme (Scheme 1) is presented after the result. 

  1. I recommend moving scheme 1 to the first section of the results to help the readers understand the protocol.

We are grateful for your feedback. You are correct, including a scheme at the beginning of the results does enhance understanding. To provide context for the image, we have added a concise description of the experimental design. 

  1. In scheme 1, please add the legends of each symbol, such as oral immunization, commercial vaccine immunization, and blood draw.

Correct, a caption has been added to the scheme.

  1. The dates in the scheme 1 are not chronological order, D3, and next D1 and again D1, The day 3 G2 received oral vaccine

Correct, we noticed the formatting error in the image and corrected the timeline.

  1. In the abstract and method, the authors indicate that G1 received silica particles without antigen on day 3. However, the scheme indicated day 2 in the scheme 1.
  2. In G4, there is no indication of day 24 oral vaccine in the scheme 1.
  3. There is no sign of commercial vaccine administration on day 24 in the scheme 1.

Regarding items "d.", "e.", and "f.": We apologize for the error, we noticed the formatting issue and have corrected the timeline.

5)    Please add a brief description of the oral vaccine and SBA-15 in the introduction

 A succinct paragraph has been added to the introduction as suggested to address the topic of SBA-15 (lines 80 – 85).

6)    Table 1. The subtitle should be “classification.”

 Thank you for the observation, the change has been made in Table 1.

7)    Tables 2, 3, and 4.  The footnote “Means followed by the same letter……(p<0.05)” is not clear. The meaning of superscripts “a” and “b” is also not explained and hard to understand the results. Please reconsider to simplify the statistical comparison.

Thank you for your suggestion. We have taken steps to address your concerns and have revised the table footnotes to enhance clarity. We regret any misunderstanding that may have arisen and have updated the table footnotes to provide a clear explanation of the meaning of the letters "a" and "b". We believe that the use of letters is widely disseminated, particularly as the Tukey test is a parametric test for comparing means, and Kruskall Wallis is a non-parametric test for comparing medians, and the results of both are typically presented using letters. This is true of many statistical analysis software programs and scientific articles, including our own recent publications (Stingelin et al., 2022; Mechler-Dreibi et al., 2021). However, we appreciate your input and welcome any additional feedback you may have to help us improve our work. Thank you for your time and valuable suggestions.

8)    Figures 1, 2, 3, 4, and 5. The figure legends are missing.  Please describe a brief protocol and Statistical analysis if performed.  Also, explain the abbreviation, e.g., Inoc in Figure 4 (Inoc=CHL?) and PUL and LTB in Figure 5. 

These figures were changed and the challenged group was removed from the results, following the suggestions of reviewer 2.

9)    Figure 2 IgA and Figure 3 IgG seem to be switched. X-axis labels are missing in both figures.

 Thank you for the suggestion. We have considered other options for graphing and have modified the parallel coordinate plots by adding boxplots. We believe that this approach provides a better representation of the data, as the boxplot contains a large amount of descriptive statistical information, such as the median, first and second quartiles, value dispersion, and outliers. As for the analysis of differences in means or medians (IgG and IgA), these differences are presented in tables included in the supplementary material, as there are many comparisons between repeated measures over time, which, in our view, makes it impractical to present all observed differences with their respective letters. If necessary, we are available to make further modifications or add letters that differentiate values in the respective graphs. The presented graphics will be available in high-definition image format.

10) Figure 4.5., Please indicate each small dots represented in the figure legends.

Thank you for the feedback. A description has been added to the tables indicating that the points represent each observation that makes up the data set, with the darker points centered above the boxplots being the outliers.

References

MECHLER-DREIBI, Marina L. et al. Oral vaccination of piglets against Mycoplasma hyopneumoniae using silica SBA-15 as an adjuvant effectively reduced consolidation lung lesions at slaughter. Scientific Reports, v. 11, n. 1, p. 22377, 2021. 

STINGELIN, Giovani Marco et al. Chemotherapeutic Strategies with Valnemulin, Tilmicosin, and Tulathromycin to Control Mycoplasma hyopneumoniae Infection in Pigs. Antibiotics, v. 11, n. 7, p. 893, 2022..

Reviewer 2 Report

Use of nanostructured silica SBA-15 as an oral vaccine adjuvant to control Mycoplasma hyopneumoniae in swine production.

 Storino et al.

 This manuscript describes the authors comparison of different vaccination regimens using a commercial M. hyopneumoniae vaccine along with the addition of oral vaccines, including the SBA-15 silica adjuvant.  The manuscript is difficult to follow as critical information seems to be missing or is included in locations that make it difficult for the reader to find.  The reference numbers sometimes do not match the references in the reference section, or the wrong papers are referenced in the text.  The writing has been done using correct but uncommon words, making it difficult for the reader to understand what information the authors are trying to convey.  The authors have performed and reported statistical analyses on everything.  Much of the statistical analyses show no significant differences, suggesting that they do not need more than a brief mention in the manuscript rather than the significant space they occupy in the results section.  Instead of going over all the individual numbers and statistical results in the text, those could represented in figures and the important statistically significant results could be covered in the text.

 Lines 20-39: The abstract has a significant presentation of the materials and method used, but lacks any indication of the results and discussion.

 Lines 95-96: This seems to be the only place where it is indicated that treatment groups came from specific litters/mothers.  Since there is a question of weight differences between groups and one mother dealing with episodes of anorexia, this information needs to be made clear as it biases the treatment groups.

 Line 113 (table 1): Subtitle line, why include (-) when it is never used in the table?

 Lines 119, 121, and elsewhere: A challenge group or challenge animals are discussed here and elsewhere in the results section.  However, this group only appears in certain sections of the results.  It is also not defined as a treatment group in materials and methods.  It likely isn’t necessary to include as a treatment group since it isn’t one, so there is no need to discuss the “challenge group” in the results section.

 Lines 156, 196, and 226 (tables 2-4): These tables are not referenced in the text and would probably work better if included in the supplementary materials.

 Lines 161, 201, and 231 (figures 1-3): These graphs may better represent your data if only the averages were plotted (I assume that is what the red stars are) and all four treatment groups are included in the same graph.  Significant differences between treatment groups can then be shown on the graph instead of in tables 2-4 and it will be much easier for the reader to understand your data and what it means.

 Lines 201 and 231 (figures 2 and 3): The plots are labeled as IgA on the Y-axis, but the figure legend says IgG for figure 2 and the opposite for figure 3.  It seems that you have your plots and figure legends mixed up.

 Line 248 (table 5): change your p-value to 0.006 from 0,006 to match the other tables.

 Line 263 (Figure 4): I can only assume that the “Inoc” group is the “challenge group” mentioned previously.  Please either include it as a treatment group everywhere, or don’t include it at all.

 Lines 283-284 (table 6): Please include information about the identification method used in table 6, similar to what you do in table 7.

 Lines 285-6: This information likely isn’t necessary (or the letters in the table) since there is no difference between treatment groups.

 Lines 319-320 (figure 5): It looks like figure 5 and table 7 should be representing the same numerical values, but the graphs in figure 5 and the numbers in table 7 do not seems to match.

 Lines 479-481: Please be careful of your description of your challenge M. hyopneumoniae strain.  You cannot say that it is different from the strain used to make the vaccine, because you never report testing for any differences.  All you can say is that you challenged with an uncharacterized field isolate.  It could be multiple M. hyopneumoniae strains or include other mycoplasma species.

 Lines 483-485: What is the difference between SBA115 (treatment G1) and the oral vaccine in treatments G2 and G4?

 Lines 498-500: Scheme 1 seems to be missing significant information, including the full numbers for the days (D0, D3, D?) and the description for the different icons that likely represent the different treatments.

 Line 514: Did you concentrate/dilute to obtain 10^5 copies/ul, or were you really that lucky to come up with that exact number?  You talk about qPCR here, but you don’t reference the M. hyopneumoniae specific primer set that you used.

 Line 558: This is a subheading and needs to be separated from the previous paragraph.

 Lines 563-567: You only inoculated 400 ml (4 flasks of 100 ml each), yet you state that you obtained a volume of 700 ml.  How was “peak orange color” determined?  Did you measure it in some way, or just eyeball each of the flasks and decided when the media was orange enough?  If you eyeballed it, you can should probably remove “the peak of”.

 Line 581: How many 1-minute rounds of sonication were performed?

 Lines 591-604: Please clarify where the SBA-15 with M. hyopneumoniae antigens was used versus the control lacking the M. hyopneumoniae antigens in your treatment groups.  It is not at all clear.

 Lines 629-637: A better scheme 1 figure would help the reader to understand your timing here.

 Line 679: You need to make it clear that all four treatment groups are cohoused in the same pen so the reader won’t wonder how you divided 15 “challenge animals) between 4 pens.  Again you really need to do a better job of describing your challenge group if it is anything more than adding the 15 animals here.

 Lines 694-696: Did you aliquot the BALF prior to freezing, or was it subject to a freeze/thaw cycle when you aliquoted it?  Did BALF collection impact lesion examination or anything else done with the lungs?

 Lines 704-705: Were the lung tissue fragments subject to a freeze thaw cycle or were they processed to standardized weight lung region prior to freezing?

 Line 712-713: Please include a brief description of your adaptation in this paragraph.

Author Response

Reviewer 2

 This manuscript describes the authors comparison of different vaccination regimens using a commercial M. hyopneumoniae vaccine along with the addition of oral vaccines, including the SBA-15 silica adjuvant.  The manuscript is difficult to follow as critical information seems to be missing or is included in locations that make it difficult for the reader to find.  The reference numbers sometimes do not match the references in the reference section, or the wrong papers are referenced in the text.  The writing has been done using correct but uncommon words, making it difficult for the reader to understand what information the authors are trying to convey.  The authors have performed and reported statistical analyses on everything.  Much of the statistical analyses show no significant differences, suggesting that they do not need more than a brief mention in the manuscript rather than the significant space they occupy in the results section.  Instead of going over all the individual numbers and statistical results in the text, those could represented in figures and the important statistically significant results could be covered in the text.

We thank you for your comments and suggestions, which were of great help in improving the paper. Changes were made in order to avoid excessive information regarding non-significant analyzes and to improve the presentation of results, maintaining data that, from our point of view, are important to understand the results and conclusions of the study, since it is an experimental study that seeks to simulate complex natural events the results can be subtle and the interpretation of the data by the reader can be impaired. In this sense, we are willing, if necessary, to make additional changes to the text to improve the paper. Regarding writing in English, we recognize that it is something limiting even in interpretation, for this reason we are willing to proceed with the submission of the text to MDPI's Language Editing Services for an extensive review of the writing prior to publication.

 Lines 20-39: The abstract has a significant presentation of the materials and method used, but lacks any indication of the results and discussion.

Thank you for the suggestion. Changes were made to the abstract and we believe they enhanced the study results (lines 34 - 39).

 Lines 95-96: This seems to be the only place where it is indicated that treatment groups came from specific litters/mothers.  Since there is a question of weight differences between groups and one mother dealing with episodes of anorexia, this information needs to be made clear as it biases the treatment groups.

A paragraph has been added to the beginning of the Results section along with a concise description of the study to contextualize the experimental design scheme added to the beginning of the paper as suggested by Reviewer #1.

 Line 113 (table 1): Subtitle line, why include (-) when it is never used in the table?

 We agree, changes were made to the table.

 Lines 119, 121, and elsewhere: A challenge group or challenge animals are discussed here and elsewhere in the results section.  However, this group only appears in certain sections of the results.  It is also not defined as a treatment group in materials and methods.  It likely isn’t necessary to include as a treatment group since it isn’t one, so there is no need to discuss the “challenge group” in the results section.

We agree with the suggestion. The sections where the results related to challenged animals were altered, as well as the graphics (Figure 4 and 5) were modified to remove the inoculated group. Additionally, a modification was made in the discussion section where the comparison of the challenged group with the others from G1 to G4 is mentioned.

 Lines 156, 196, and 226 (tables 2-4): These tables are not referenced in the text and would probably work better if included in the supplementary materials.

We agree with the suggestion. We have transferred the mentioned tables to the supplementary material section.

 Lines 161, 201, and 231 (figures 1-3): These graphs may better represent your data if only the averages were plotted (I assume that is what the red stars are) and all four treatment groups are included in the same graph.  Significant differences between treatment groups can then be shown on the graph instead of in tables 2-4 and it will be much easier for the reader to understand your data and what it means.

Thank you for your suggestion. We have made changes to the mentioned graphs by including parallel coordinate boxplots that show the median values at each time point. We believe it is particularly interesting to observe how the data varies over time by relating individual observations to identify trends of increasing or decreasing values among some individuals or the group as a whole. We appreciate the input of the reviewer and we are available to make further changes if needed. Additionally, we would like to inform you that the graphs will be provided in high definition so that they can be easily viewed by readers on the website.

 Lines 201 and 231 (figures 2 and 3): The plots are labeled as IgA on the Y-axis, but the figure legend says IgG for figure 2 and the opposite for figure 3.  It seems that you have your plots and figure legends mixed up.

As mentioned above, the graphics were changed. Thank you for your suggestion

 Line 248 (table 5): change your p-value to 0.006 from 0,006 to match the other tables.

The change was carried out as requested.

 Line 263 (Figure 4): I can only assume that the “Inoc” group is the “challenge group” mentioned previously.  Please either include it as a treatment group everywhere, or don’t include it at all.

Right, with the modification in the description of the results and removal of information regarding the inoculated animals, the graphics and tables were also modified.

 Lines 283-284 (table 6): Please include information about the identification method used in table 6, similar to what you do in table 7.

 Agreed. The table caption was modified as requested.

 Lines 285-6: This information likely isn’t necessary (or the letters in the table) since there is no difference between treatment groups.

 Thank you for the observation, indeed. The modification was carried out on the table.

 Lines 319-320 (figure 5): It looks like figure 5 and table 7 should be representing the same numerical values, but the graphs in figure 5 and the numbers in table 7 do not seems to match.

Thank you for your observation. To ensure both the results of the graph and the analyses, they were checked and found to be correct. Regarding the image, it is important to note that due to the influence of extreme values and high variability, as we can observe in the outliers in the boxplot, although it is a good tool for visual analysis of the results, it may have limitations in the observation of measures of central tendency, given the flattening of the image near the x-axis.

 Lines 479-481: Please be careful of your description of your challenge M. hyopneumoniae strain.  You cannot say that it is different from the strain used to make the vaccine, because you never report testing for any differences.  All you can say is that you challenged with an uncharacterized field isolate.  It could be multiple M. hyopneumoniae strains or include other mycoplasma species.

We agree with the observation. Parts of the text containing the term "heterologous strain" referring to the field isolate have been replaced with "field strain" and uncategorized field strain.

 Lines 483-485: What is the difference between SBA115 (treatment G1) and the oral vaccine in treatments G2 and G4?

 A modification was made to the section, adding more information regarding the G1 protocol, making it clear that it does not contain M. hyopneumoniae antigens in its composition.

 Lines 498-500: Scheme 1 seems to be missing significant information, including the full numbers for the days (D0, D3, D?) and the description for the different icons that likely represent the different treatments.

 Correct observation. In addition to the modification of the formatting error, scheme 1 was moved to the beginning of the results section, as suggested by reviewer #1.

 Line 514: Did you concentrate/dilute to obtain 10^5 copies/ul, or were you really that lucky to come up with that exact number?  You talk about qPCR here, but you don’t reference the M. hyopneumoniae specific primer set that you used.

 Regarding the quantification of M. hyopneumoniae, our aim was to achieve a concentration of at least 10^5 copies, as it was reported in the literature that this concentration would be a minimum infective dose, although there are still few reports on the subject. As the intention was to ensure the infection of the females, the experimental period would only be initiated if we had an inoculum at these concentrations. We based our approach on something similar to what was done by Robins et al. (2019), although we were not interested in cultivate M. hyopneumoniae, as the cultures media are selective and would be far from the reality of M. hyopneumoniae dissemination in the field, along with other bacterial microorganisms. Fortunately, in the first attempt to produce the lung homogenate, a concentration of 10^5 copies was obtained, allowing the study to be initiated with the challenge of pregnant females and CHL piglets.

 Line 558: This is a subheading and needs to be separated from the previous paragraph.

 A modification was made to the section.

 Lines 563-567: You only inoculated 400 ml (4 flasks of 100 ml each), yet you state that you obtained a volume of 700 ml.  How was “peak orange color” determined?  Did you measure it in some way, or just eyeball each of the flasks and decided when the media was orange enough?  If you eyeballed it, you can should probably remove “the peak of”.

A change was made regarding the cultivation, as indeed, two cultivation steps were performed resulting in approximately 400 mL each. As it is a consolidated method, the observation of the color with the display of the orange color results in approximate concentrations, generally around 10^6 and 10^7, which are ideal for vaccine production, as with the display of a yellowish color, after the "peak" of multiplication, a process of bacterial death begins, which can negatively influence the quality of the vaccine produced. Therefore, although quantification was performed by successive passages, they were carried out after cultivation. We have made the suggested change by removing the term "peak" in the passage.

 Line 581: How many 1-minute rounds of sonication were performed?

 Thank you for the observation. A change was made to the sentence. In total, three rounds were performed.

 Lines 591-604: Please clarify where the SBA-15 with M. hyopneumoniae antigens was used versus the control lacking the M. hyopneumoniae antigens in your treatment groups.  It is not at all clear.

 Thank you for the feedback, indeed it was not clear the difference in the production of the compounds. We have amended the paragraph in question to enhance understanding.

 Lines 629-637: A better scheme 1 figure would help the reader to understand your timing here.

 Correct observation. We have adjusted the timeline in scheme 1.

 Line 679: You need to make it clear that all four treatment groups are cohoused in the same pen so the reader won’t wonder how you divided 15 “challenge animals) between 4 pens.  Again you really need to do a better job of describing your challenge group if it is anything more than adding the 15 animals here.

 Thank you for the observation. The paragraphs in section 4.8 have been revised to improve understanding regarding the challenge during the nursery phase.

 Lines 694-696: Did you aliquot the BALF prior to freezing, or was it subject to a freeze/thaw cycle when you aliquoted it?  Did BALF collection impact lesion examination or anything else done with the lungs?

Thanks for the question. As this is a collection carried out at the time of necropsies, in order to avoid external contamination, the content in the sterile pipettes was dispensed directly into DNAse and RNAse free 15 mL falcon tubes and immediately frozen at -80ºC and thawed and aliquoted right before to DNA extraction. BALF samples were collected after assessing the extent of lung lesions and previously collecting fragments for qPCR, avoiding PBS leakage due to the cut performed to obtain the tissue sample.

 Lines 704-705: Were the lung tissue fragments subject to a freeze thaw cycle or were they processed to standardized weight lung region prior to freezing?

Due to the size of the fragment, 0.05g, larger portions of the lung of approximately 5g were selected and stored in Whirl Pak DNAse and RNAse free and frozen at -80ºC. Before the DNA extractions, the fragments were thawed and internal portions of 0.05g were selected in a clean and disinfected laboratory environment to immediately start the extraction protocol.

 Line 712-713: Please include a brief description of your adaptation in this paragraph.

As requested, information on the adaptations made has been added to the paragraph (lines 776-785).

ROBBINS RC, BETLACH AM, MONDRAGON-EVANS MR, et al. Development of a herd-specific lung homogenate for exposure to Mycoplasma hyopneumoniae under field conditions. J Swine Health Prod, 2019.

Round 2

Reviewer 1 Report

I appreciate the authors addressed most of my concerns. As the authors mentioned, a few additional comments will need to be addressed.   

3) The lacking negative control will make the statistical comparison. Therefore, I recommend the authors add a note described at the beginning of the results.

7) Regarding the issue of "Means followed by the same letter “a” and “b” on the same row are not significantly different by Tukey’s parametric test (p < 0.05)", could the authors mark the values which are significantly different, instead of "not significantly different"? Also, G3 could be a baseline control that can be used for the comparator since it received only the commercial vaccine. 

Author Response

Reviewer comment: I appreciate the authors addressed most of my concerns. As the authors mentioned, a few additional comments will need to be addressed. We would like to thank you for your attention to our paper. The suggestions from the first round were essential to improve the manuscript and we hope to have responded in the best way to the additional comments. We remain at your disposal to make any additional changes as necessary.

3) The lacking negative control will make the statistical comparison. Therefore, I recommend the authors add a note described at the beginning of the results.

As suggested, a modification was made (lines 103 to 109) to specify that all study groups received the vaccine via intramuscular injection, and to report that Group 3 was the only group that received only the commercial intramuscular vaccine.

7) Regarding the issue of "Means followed by the same letter “a” and “b” on the same row are not significantly different by Tukey’s parametric test (p < 0.05)", could the authors mark the values which are significantly different, instead of "not significantly different"? Also, G3 could be a baseline control that can be used for the comparator since it received only the commercial vaccine.

Changes have been made to tables 5, S1, S3, and S5, and footnotes have been updated to indicate significant differences. As a result of the changes, repeated letters have been removed, leaving the letter only on significant differences. Additionally, modifications have been made to tables 6 and 7. While non-numerical markings could be suitable for isolated cases of significant differences, as seen in table 5, they become unfeasible in cases such as table S3, where four differences are observed among group values, making letters essential to determinate the statistical difference. Therefore, we have aimed to maintain the letter convention, omitting repetitions to emphasize significant differences, as suggested. The reviewer is correct regarding the use of G3 as a baseline, and we appreciate the opportunity to clarify that in Dunn/Kruskall Wallis tests, as well as in Tukey's pairwise comparisons of groups, the median/mean ranks of each group are compared to the median/mean ranks of every other group. Consequently, in all analyses for comparison among groups, individualized comparison p-values of G3 with the others were obtained, and when significant trends or p-values were observed, they were presented in the text or tables.